# EPS3D: End-to-End Feed-Forward 3D Panoptic Segmentation

**Runsong Zhu** [1 2]  **Jiaxin Guo** [1 2]  **Xiaoyang Guo** [3 †]  **Zhengzhe Liu** [4 †]  **Ka-Hei Hui** [5]  **Wei Yin** [3]  **Kai Chen** [1]
**Wei Chen** [1 2]  **Weiqiang Ren** [3]  **Yunhui Liu** [1 2]  **Pheng-Ann Heng** [1 2]  **Chi-Wing Fu** [1 2]

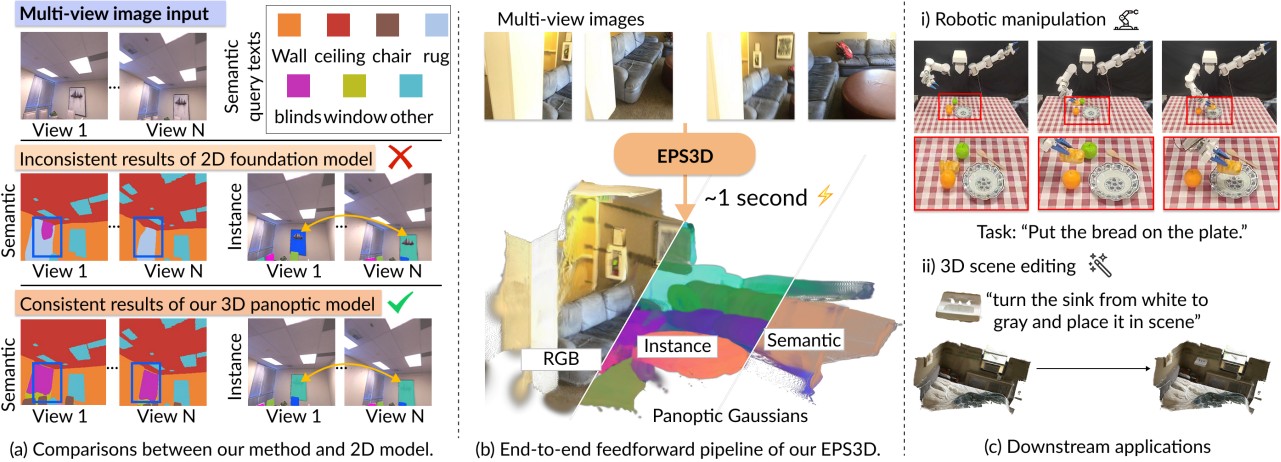

*Figure 1.* (a) While 2D foundation models struggle with view inconsistency, our method, EPS3D, can provide effective 3D open-vocabulary panoptic segmentation and can render accurate and view-consistent 2D segmentation across views. (We visualize only "chair" and "paint" instance masks for simplicity.) (b) From multi-view images, we rapidly provide 3D panoptic segmentation via 3D panoptic Gaussian reconstruction. (c) EPS3D is able to further facilitate various downstream tasks like robotic manipulation and 3D scene editing.

## Abstract

This paper introduces EPS3D, a new end-to-end feed-forward framework for open-vocabulary 3D panoptic segmentation. Unlike existing methods relying on additional preprocessing, we design an end-to-end architecture, with a distillation-based training strategy on diverse 3D scenes to predict 3D-aware semantic and instance features from multi-view images, improving 3D consistency and avoiding error accumulation. We further propose a mutual enhancement module to enforce inherent semantic-instance consistency. By aligning semantics within instances (Ins2Sem) and refining instance features with semantic guidance (Sem2Ins), we achieve more coherent 3D scene understanding. Ultimately, EPS3D outperforms SOTA baselines on two benchmarks (e.g., +13% mIoU for semantics on Replica) with high efficiency (e.g., 1s per scene), supporting tasks like robotic manipulation and 3D scene editing.

## 1. Introduction

Open-vocabulary 3D panoptic segmentation (OV3DPS) is a challenging task. It requires not only assigning unrestricted semantic categories and instance identities in a 3D scene but also enforcing 3D-consistent predictions across the scene. Hence, OV3DPS has great value in supporting diverse physical AI applications in robotics, embodied AI, and VR/AR.

However, accurate and robust OV3DPS is too challenging to obtain, due to limited 3D dataset and labor-intensive manual 3D annotations. Existing methods (Kobayashi et al., 2022; Zhou et al., 2024; Cen et al., 2024; Zhu et al., 2025; Shi et al., 2024; Jun-Seong et al., 2025; Ji et al., 2025; Ye et al., 2023; Wu et al., 2024b; Li et al., 2024; Zhu et al., 2025) suggest lifting the results of 2D foundation models (e.g., CLIP (Radford et al., 2021) for semantic understanding and

---
[1]The Chinese University of Hong Kong, HK SAR, China [2]Hong Kong Centre for Logistics Robotics, HK SAR, China [3]Horizon Robotics, China [4]Lingnan University, HK SAR, China [5]Autodesk AI Lab, Toronto, Canada. Correspondence to: Xiaoyang Guo <xiaoyang.guo1995@gmail.com>, Zhengzhe Liu <zhengzhe-liu@ln.edu.hk>.

*Proceedings of the 43rd International Conference on Machine Learning*, Seoul, South Korea. PMLR 306, 2026. Copyright 2026 by the author(s).

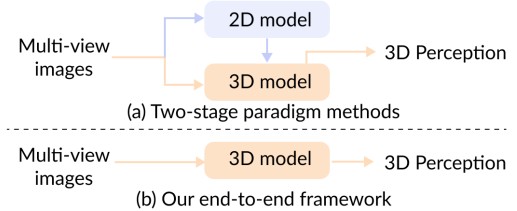

*Figure 2.* Comparisons between recent SOTA methods (Fan et al., 2024; Sun et al., 2025) and our method, EPS3D.

SAM (Kirillov et al., 2023) for instance-level prediction) into a 3D scene (e.g., 3D radiance field). Yet, 2D results inferred from individual views typically suffer from view imperfection and inconsistent semantic predictions. Also, 2D instance segmentation often fails to maintain consistent object identities across views; see Fig. 1 (a). To address the challenges, these methods adopt per-scene optimization fusion solution, by combining 2D and 3D pre-processing such as 3D Gaussian splatting optimization and 2D segmentation extraction. Though significant progress has been made, the per-scene optimization fails to offer scene-level robustness. Also, it is too time costly for real-time applications.

More recently, some methods (Fan et al., 2024; Sun et al., 2025) attempt to understand 3D scenes with feed-forward 3D reconstruction models. However, despite their improved efficiency, these approaches follow a two-stage design: first extracting per-view semantic features using a pretrained 2D model, then using a feed-forward 3D network mainly to fuse the independently inferred features across views; see Fig. 2. Since the intermediate 2D features are view-dependent and thus inconsistent, the subsequent multi-view fusion is prone to error accumulation. Also, they typically do not leverage object-level structural cues and focus on semantic-level understanding, often leading to blurred boundaries with limited ability to support instance-level, object-centric applications (e.g., editing and robotic interaction). So, we raise this question: can we have an effective and efficient end-to-end 3D panoptic segmentation method that eliminates **error accumulation**, while jointly supporting **accurate semantic and object-level predictions in 3D**?

In this work, we propose EPS3D, the first end-to-end feed-forward framework for open-vocabulary *3D panoptic* segmentation from unposed multi-view images; see Fig. 2. EPS3D directly predicts a unified set of 3D panoptic Gaussians that encode geometry, appearance, and panoptic attributes, enabling 3D understanding and efficient novel-view rendering. During training, we leverage 2D foundation models only as teachers to provide distillation supervision across diverse scenes, avoiding reliance on 3D panoptic annotations.[1] Based on the end-to-end design, we effectively reduce the error accumulation in previous methods.

---

[1]The 2D foundation models are not part of our method but serve solely as external teachers that provide supervision signals.

More importantly, our end-to-end formulation naturally calls for jointly exploiting semantic and instance cues; thus, we introduce a semantic-instance mutual enhancement module to couple the two predictions. Our key insight is that semantic and instance cues are strongly complementary: instance features capture object-level structure and boundary cues that can sharpen semantic predictions, while semantic features provide category-level context that stabilizes instance grouping. Accordingly, we propose two coupled components: (i) Instance2Semantic (Ins2Sem), which enforces semantic consistency among Gaussians that belong to the same instance, and (ii) Semantic2Instance (Sem2Ins), which uses semantic guidance to refine instance features for more robust and consistent instance segmentation. By actively integrating semantic and instance features, we effectively promote more coherent 3D panoptic prediction.

We evaluate our method based on two standard benchmarks, including ScanNet (Dai et al., 2017) and Replica (Straub et al., 2019) for open-vocabulary semantic and instance segmentation. The quantitative and qualitative results demonstrate that our method achieves significant improvements (e.g., +13% mIoU for semantics on Replica) over existing baselines with high inference efficiency (1s per scene). Further, we demonstrate that our EPS3D can facilitate a variety of downstream applications, including robotic manipulation and 3D scene editing; see Fig. 1 (c).

Our major contributions are summarized as follows:

- Unified Framework: We propose EPS3D, the first end-to-end feed-forward framework for open-vocabulary 3D panoptic segmentation from images that avoids error accumulation and boosts 3D consistency.

- Coherent Prediction: By actively enforcing semantic-instance consistency, our method produces significantly more coherent 3D panoptic results.

- Superior Performance & Application: Our approach outperforms baselines on two standard benchmarks with high efficiency, further supporting tasks such as robotic manipulation and scene editing.

## 2. Related Work

### 2.1. Differentiable 3D Representation

Radiance fields have emerged as a powerful differentiable representation for supporting 3D scene reconstruction with diverse properties such as geometry, color, semantics, and instance information. Neural Radiance Fields (NeRF) (Mildenhall et al., 2021) model the radiance field using neural networks composed of MLPs, enabling photorealistic novel view synthesis. Subsequent works focus on improving the efficiency of NeRF by introducing explicit 3D

structures, such as voxel grids (Chen et al., 2022; Liu et al., 2020) and hash grids (Müller et al., 2022). More recently, 3D Gaussian Splatting (3D-GS) (Kerbl et al., 2023; Xu et al., 2024; Liang et al., 2024; Zhang et al., 2024b; Cheng et al., 2024; Huang et al., 2024a; Yu et al., 2024; Zhang et al., 2024a; Kulhanek et al., 2024; Jiang et al., 2025; Sun et al., 2025) is proposed as an alternative representation by modeling the radiance field as a set of explicit 3D Gaussian points. This approach supports splatting-based rendering (Kopanas et al., 2021), which is highly efficient, significantly enhancing its potential for real-time applications. Given these advantages, we adopt 3D-GS as the backbone representation in our end-to-end 3D panoptic segmentation framework.

### 2.2. 3D Scene Understanding

3D scene segmentation has made significant progress in recent years, empowered by 2D vision foundation models (VFMs) (e.g., CLIP (Radford et al., 2021), LSeg (Li et al., 2022), DINO (Caron et al., 2021)). To address the view inconsistency issues in VFM predictions, these methods propose to fuse dense VFM features from multi-view images into a shared continuous radiance field representation, enabling high-resolution novel-view synthesis for feature alignment and downstream tasks. Early approaches (Kerr et al., 2023; Qin et al., 2024; Zhu et al., 2025; Ye et al., 2023; Wu et al., 2024b; Jun-Seong et al., 2025; Ji et al., 2025; Li et al., 2024; Zhu et al., 2026; Bhalgat et al., 2024; Engelmann et al., 2024; Guo et al., 2024; Zhu et al., 2024) typically rely on per-scene optimization. While promising, this strategy is computationally expensive, limiting its applicability for real-world scenarios.

Recent baselines (Fan et al., 2024; Sun et al., 2025) attempt to integrate 3D scene understanding into feed-forward 3D reconstruction models (e.g., Dust3R (Wang et al., 2024b), VGGT (Wang et al., 2025a)); however, these methods still follow a two-stage paradigm. For instance, LSM (Fan et al., 2024) first uses 2D VFMs (Li et al., 2022) to extract semantic feature maps per view, then fuses them via Point-Transformer (Wu et al., 2024a). Similarly, the concurrent work Uni3R (Sun et al., 2025) employs the VGGT (Wang et al., 2025a) architecture to fuse features extracted by 2D VFMs (Li et al., 2022). Despite improved efficiency, this two-stage paradigm is prone to error accumulation and lacks robustness against 2D inconsistencies. Besides, since the approach focuses solely on semantic prediction, it lacks object-level understanding required for many downstream tasks. In this work, we address these limitations by introducing a new end-to-end 3D panoptic segmentation model that directly infers semantics, instances, and geometry from raw inputs, achieving significantly improved performance with scene-level robustness and high computational efficiency.

### 2.3. Feed-Forward 3D Reconstruction

3D reconstruction methods such as NeRF (Mildenhall et al., 2021) and 3DGS (Kerbl et al., 2023) deliver high-fidelity renderings. Yet, the computation is time-consuming and the trained model is per-scene. Recent approaches have significantly accelerated reconstruction for both Gaussian-based (Jiang et al., 2025; Charatan et al., 2024; Chen et al., 2024; Guo et al., 2025) and point-map-based (Shen et al., 2025; Wang et al., 2025b;a; 2024b; Xie et al., 2026; Cheng et al., 2026; Deng et al., 2025) frameworks by exploring the feed-forward 3D reconstruction models. However, these methods predominantly concentrate on geometry and appearance, neglecting high-level semantic and instance understanding. This paper presents a feed-forward solution, not only embedding both semantics and instances into a 3D representation but also recovering high-quality geometry and appearance from multi-view images, setting a new state-of-the-art for 3D scene understanding.

## 3. Method

### 3.1. Problem Definition

Given a collection of $N$ unposed RGB images $\mathcal{C} = \{C_i\}_{i=1}^N$ of a static 3D scene, where $C_i \in \mathbb{R}^{H \times W \times 3}$, our goal is to train a feed-forward model to provide 3D panoptic understanding by simultaneously predicting scene geometry, appearance, along with their semantics and instance in a unified 3D panoptic Gaussian representation $\mathcal{G}$:

$$\mathcal{G} = \{ \underbrace{(I_g, S_g)}_{\text{semantic and instance}}, \underbrace{(\boldsymbol{\mu}_g, \sigma_g, \boldsymbol{r}_g, \boldsymbol{s}_g, \boldsymbol{c}_g)}_{\text{geometry and appearance}} \}_{g=1}^G, \quad (1)$$

where the geometry and appearance of each Gaussian is parameterized by its center $\boldsymbol{\mu} \in \mathbb{R}^3$, opacity $\sigma \in \mathbb{R}^+$, rotation quaternion $\boldsymbol{r} \in \mathbb{R}^4$, anisotropic scaling $\boldsymbol{s} \in \mathbb{R}^3$, and color embedding $\boldsymbol{c} \in \mathbb{R}^{3 \times (k+1)^2}$ represented by spherical harmonic coefficients of degree $k$ (Kerbl et al., 2023). In addition, the semantic and instance of each Gaussian is parametrized by the Gaussian-wise text-aligned semantic feature (Radford et al., 2021) $S_g \in \mathbb{R}^{D_S}$ and the Gaussian-wise instance feature $I_g \in \mathbb{R}^{D_I}$, where $D_S$ is set to 512 as the CLIP feature dimension (Radford et al., 2021) and $D_I$ is set to 32 by default. Mathematically, our model $f_\theta$ learns the mapping:

$$f_\theta : \{C_i\}_{i=1}^N \longmapsto \mathcal{G}. \quad (2)$$

Following the color rendering in the standard 3D Gaussians (Kerbl et al., 2023), our panoptic Gaussian is able to additionally render the semantic map and instance map from the novels, supporting various applications, including scene editing and robotic manipulation (see Sec. 4.4).

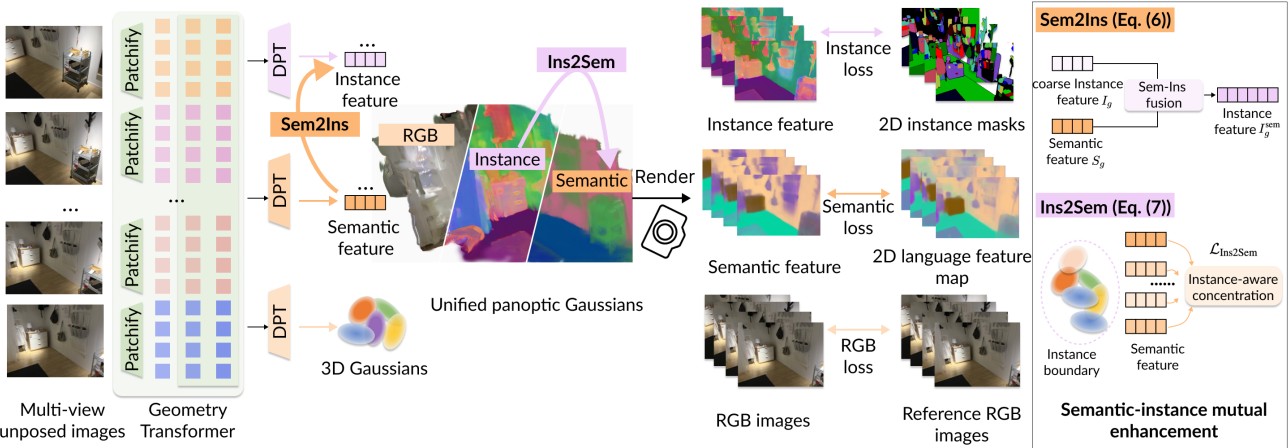

*Figure 3.* Overview of EPS3D. Given multi-view images as input, EPS3D provides 3D panoptic segmentations by predicting unified panoptic 3D Gaussians in a feed-forward pass, supporting novel view RGB, semantic and instance feature map rendering. With our end-to-end framework, we further introduce semantic-instance mutual enhancement learning module (i.e., Semantic2Instance (Sem2Ins) learning and Instance2Semantic (Ins2Sem) enhancement) to integrate the complementary knowledge of semantic and instance understanding.

## 3.2. End-to-End Feed-Forward 3D Panoptic Segmentation Framework

Fig. 3 illustrates the overall framework. Our pipeline first encodes a set of unposed multi-view images into high-dimensional feature representations using a geometry transformer architecture (Wang et al., 2025a; Jiang et al., 2025). These features are subsequently decoded into 3D panoptic Gaussians, which include standard 3D Gaussian parameters alongside panoptic parameters (i.e., semantic and instance features) to enable 3D panoptic segmentation (see Sec.3.3). Using these 3D panoptic Gaussians, we render RGB, semantic, and instance images, and impose supervision on the semantic and instance feature images through two specialized supervisions (see Sec.3.4.1). Furthermore, we propose a semantic-instance mutual enhancement learning module to encourage inherent consistency between semantic and instance features, thereby facilitating effective 3D panoptic segmentation training (see Sec.3.4.2). Finally, we introduce the inference for our 3D open-vocabulary panoptic segmentation (see Sec.3.4.3).

## 3.3. End-to-End Architecture

Given the input multi-view images, we first extract image tokens using the geometry transformer backbone, and then decode them with a Gaussian head to regress the geometry and appearance parameters, together with two heads to predict Gaussian-wise semantic and instance features.

**Geometry transformer.** Based on VGGT (Wang et al., 2025a), we first patchify each image $C_i$ into tokens. These initial tokens are fed into an $L$-layer transformer with self-attention and cross-attention to produce the 3D-aware aggregated tokens $\hat{t}^i$ for each image $C_i$.

**Gaussian head.** We employ a dual-head architecture based on the DPT decoder (Ranftl et al., 2021) to regress the Gaussian primitives, following common practice (Jiang et al., 2025; Sun et al., 2025). The first DPT head $F_D$ processes image tokens $\hat{t}^i$ to generate per-pixel depth maps $D_i$, which can be back-projected to the 3D space as the Gaussian centers $\{\boldsymbol{\mu}_g\}_{g=1}^G$ of the Gaussians. Meanwhile, another separate DPT head is utilized to predict the remaining attributes: opacity $\sigma_g$, orientation $\boldsymbol{r}_g$, scale $\boldsymbol{s}_g$, and SH color coefficients $\boldsymbol{c}_g$.

**Semantic and instance heads.** In our EPS3D, we propose to directly predict the semantic feature and instance feature directly from the 3D-aware geometry transformer layers. To align with the predicted pixel-wise Gaussian, we employ the standard DPT architectures $F_I$ and $F_S$, which operate on the extracted image token $\hat{t}^i$ to predict the text-aligned semantic feature $S_g$ (i.e., CLIP (Li et al., 2022)) as well as the instance features $I_g$. These features jointly offer a complementary panoptic understanding of the 3D scene. Mathematically, the process is formulated as:

$$\{I_g, S_g\} = F_I(\hat{t}^i), F_S(\hat{t}^i). \tag{3}$$

Particularly, we aim to obtain multi-view consistent semantic and instance features prediction $\{I_g, S_g\}$ in an end-to-end manner, complementing the prediction of unified 3D panoptic Gaussians $\mathcal{G}$.

**Discussion.** Existing feed-forward baselines (e.g., LSM (Fan et al., 2024), Uni3R (Sun et al., 2025)) typically precompute per-view semantic features using a 2D model (e.g., LSeg (Li et al., 2022)) and then fuse them with a 3D network, as shown in Fig. 2. Because these features are extracted independently for each view, cross-view

inconsistencies can arise and subsequently accumulate during 3D fusion. In contrast, EPS3D directly takes multi-view RGB images as input and predicts a unified 3D representation end-to-end, which encourages multi-view consistency during feature extraction and decoding.

### 3.4. 3D Panoptic Gaussian Training with Semantic-Instance Mutual Enhancement

We train EPS3D by optimizing the network parameters to predict a 3D panoptic Gaussian representation $\mathcal{G}$ that jointly encodes geometry/appearance and panoptic attributes (semantic and instance features). Specifically, we supervise the geometry and appearance parameters using the standard RGB rendering loss with common regularization terms (Jiang et al., 2025; Ye et al., 2024). For the panoptic attributes, we adopt a distillation-based supervision from 2D foundation models, and further introduce a semantic–instance mutual enhancement module to explicitly encourage consistency between semantic and instance predictions.

#### 3.4.1. BASIC TRAINING FRAMEWORK

Our basic training uses two losses to supervise semantic and instance prediction.

**Text-aligned semantic loss.** Specifically, to optimize the semantic head $F_S$, intuitively, we can maximize the similarity between the rendered text-aligned semantic feature $S^i$ and the semantic feature $\hat{S}^i$ from the 2D model (Li et al., 2022). Mathematically, we choose the cosine similarity to measure the similarity, and the semantic loss function can be formulated as:

$$\mathcal{L}_{sem} = 1 - \frac{\hat{S}^i \cdot S^i}{\|\hat{S}^i\|\|S^i\|}. \tag{4}$$

By applying the above loss function, we enforce the semantic heads to directly predict the text-aligned semantic features from the geometry transformer tokens.

**Instance contrastive loss.** Compared with the semantic feature, due to the issue of inconsistent instance ID, the supervision for instance feature has to be invariant to the permutation of the 2D segmentation ID in segmentation masks $\mathcal{K}_i$ extracted by 2D instance model (i.e., SAM (Kirillov et al., 2023)). To this end, we incorporate the single-view contrastive learning in our end-to-end framework to solve the issue. Specifically, we also first render the instance feature and apply the InfoNCE loss to conduct the contrastive learning. Mathematically, the loss term is formulated as:

$$\mathcal{L}_{ins} = -\frac{1}{|\Omega|} \sum_{\Omega_j \in \Omega} \sum_{u \in \Omega_j} \log \frac{\exp\left(\text{sim}\left(I_u, \overline{I}_j\right)\right)}{\sum_{\Omega_I \in \Omega} \exp\left(\text{sim}\left(I_u, \overline{I}_l\right)\right)}, \tag{5}$$

where $I_u$ denotes the rendered instance feature at each pixel $u$, similarity kernel function sim uses the dot product operation here and $\Omega$ is the set of pixel samples. In particular, $\Omega_j, \Omega_l$ denotes the pixel samples with the same instance ID $j, l$ according to the 2D instance segmentation $\mathcal{K}_i$, $\overline{I}_j$ and $\overline{I}_l$ represent the mean instance features (centroids) for $\Omega_j$ and $\Omega_l$, respectively. By minimizing this loss, the instance head learn to produce discriminative and view-consistent features that capture 3D instance-level information.

#### 3.4.2. SEMANTIC-INSTANCE MUTUAL ENHANCEMENT

While the basic training scheme provides effective supervision for semantic and instance features, it optimizes the two branches with separate objectives, resulting in largely independent representations. However, semantic and instance cues are inherently correlated: semantic features encourage category-level consistency, whereas instance features capture object-level grouping and boundary structure. To better exploit this complementarity, we introduce a scheme that explicitly couples the two representations during training. Specifically, we refine instance features using semantic guidance (Sem2Ins) and regularize semantic features using instance-aware neighborhoods (Ins2Sem), thereby promoting more coherent panoptic predictions.

**Semantic-aware instance (Sem2Ins) learning.** On the one hand, semantic information provides effective guidance for fine-grained instance-level feature prediction, which motivates us to design the semantic-aware instance (Sem2Ins) learning module to simplify instance feature learning. To this end, we propose to fuse the predicted semantic feature $S_G$ with the initial predicted instance feature $I_g$ to perform semantic-aware instance feature refinement, producing more accurate instance features $I_g^{sem}$. Technique-wise, we first project the initial instance feature $I_g$ and the semantic feature $S_G$ into a new feature space to perform a concatenation operation, followed by further feature fusion. Mathematically, we formulate the learning process as:

$$\{I_g^{\text{sem}}\} = F_{\text{fusion}}\left(\text{concat}(F_{\text{proj1}}(I_g), F_{\text{proj2}}(S_g))\right). \tag{6}$$

Here, $F_{\text{fusion}}$, $F_{\text{proj1}}$, $F_{\text{pro2}}$ are semantic-instance (Sem-Ins) fusion layer, initial instance feature projection layer, and semantic feature projection layer, respectively. The resulting semantic-refined instance feature $I_g^{\text{sem}}$ serves as the final instance attribute for each 3D Gaussian. During the rendering, this attribute is splatted to produce the 2D instance feature map, which is then directly supervised by the instance contrastive loss $\mathcal{L}_{\text{ins}}$ (Eq. (5)).

**Instance2Semantic (Ins2Sem) enhancement.** On the other hand, the instance feature provides discriminative boundary information, which can be used to enhance semantic quality. Particularly, 3D Gaussians that belong to

*Table 1.* Quantitative comparison for 2-views settings on the ScanNet (Dai et al., 2017). The metrics are categorized into semantic and instance quality across context and novel views. "FF." refers to the feed-forward method. "Sem." and "Inst." denote Semantic and Instance tasks, respectively. We report class-wise intersection over union (mIoU), average pixel accuracy (Acc.), cross-view intersection over union (mIoU), and F-score (F-sc.). Besides, the reconstruction time is measured on a single NVIDIA A800 GPU.

| Method | Perception Model | FF. | Sem. | Ins. | Time | Context Sem. | | Context Inst. | | Novel Sem. | | Novel Inst. | |
|---|---|---|---|---|---|---|---|---|---|---|---|---|---|
| | | | | | | mIoU↑ | Acc.↑ | mIoU↑ | F-sc.↑ | mIoU↑ | Acc.↑ | mIoU↑ | F-sc.↑ |
| LSeg (Li et al., 2022) | 2D | ✓ | ✓ | | 0.2s | 0.4701 | 0.7891 | - | - | - | - | - | - |
| SAM (Kirillov et al., 2023) | 2D | ✓ | | ✓ | 1.6s | - | - | 0.3659 | 0.1150 | - | - | 0.3363 | 0.1049 |
| NeRF-DFF (Kobayashi et al., 2022) | Two-stage 3D | | ✓ | | 1min | 0.4540 | 0.7173 | - | - | 0.4037 | 0.6755 | - | - |
| Feature-3DGS (Zhou et al., 2024) | Two-stage 3D | | ✓ | | 18min | 0.4453 | 0.7276 | - | - | 0.4223 | 0.7174 | - | - |
| Unified-Lift (Zhu et al., 2025) | Two-stage 3D | | | ✓ | 5min | - | - | 0.1441 | 0.2009 | - | - | 0.1917 | 0.3118 |
| LSM (Fan et al., 2024) | Two-stage 3D | ✓ | ✓ | | 0.43s | 0.5034 | 0.7740 | - | - | 0.5078 | 0.7686 | - | - |
| Uni3R (Sun et al., 2025) | Two-stage 3D | ✓ | ✓ | | 0.85s | 0.5233 | 0.8188 | - | - | 0.5336 | 0.8173 | - | - |
| EPS3D (ours) | End-to-end 3D | ✓ | ✓ | ✓ | 0.73s | **0.6323** | **0.8465** | **0.4147** | **0.4552** | **0.6432** | **0.8555** | **0.4227** | **0.4387** |

*Table 2.* Quantitative comparison for multi-view (i.e., 8-views) settings on ScanNet and replica dataset. "Sem." and "Inst." denote Semantic and Instance tasks, respectively. For semantic metric, following the common practices in (Fan et al., 2024; Sun et al., 2025), we report class-wise intersection over union (mIoU) and average pixel accuracy (Acc.). For instance metric, we report cross-view intersection over union (mIoU) and F-score (F-sc.), following Unified-Lift (Zhu et al., 2025).

| Method | ScanNet (Dai et al., 2017) | | | | | | | | Replica (Straub et al., 2019) | | | | | | | |
|---|---|---|---|---|---|---|---|---|---|---|---|---|---|---|---|---|
| | Context Sem. | | Context Inst. | | Novel Sem. | | Novel Inst. | | Context Sem. | | Context Inst. | | Novel Sem. | | Novel Inst. | |
| | mIoU↑ | Acc.↑ | mIoU↑ | F-sc.↑ | mIoU↑ | Acc.↑ | mIoU↑ | F-sc.↑ | mIoU↑ | Acc.↑ | mIoU↑ | F-sc.↑ | mIoU↑ | Acc.↑ | mIoU↑ | F-sc.↑ |
| LSeg (Li et al., 2022) | 0.4843 | 0.7977 | - | - | 0.4513 | 0.7727 | - | - | 0.3614 | 0.8210 | - | - | 0.4038 | 0.8224 | - | - |
| SAM (Kirillov et al., 2023) | - | - | 0.2706 | 0.1002 | - | - | 0.2792 | 0.0910 | - | - | 0.2935 | 0.1489 | - | - | 0.2781 | 0.1077 |
| Feature-3DGS (Zhou et al., 2024) | 0.2341 | 0.6846 | - | - | 0.2239 | 0.6702 | - | - | 0.3416 | 0.8173 | - | - | 0.3260 | 0.7932 | - | - |
| Unified-Lift (Zhu et al., 2025) | - | - | 0.1456 | 0.1248 | - | - | 0.1876 | 0.1304 | - | - | 0.1046 | 0.1321 | - | - | 0.2264 | 0.1537 |
| LSM (Fan et al., 2024) | 0.3366 | 0.7398 | - | - | 0.3351 | 0.7350 | - | - | 0.2801 | 0.6331 | - | - | 0.2874 | 0.6787 | - | - |
| Uni3R (Sun et al., 2025) | 0.4885 | 0.8022 | - | - | 0.5215 | 0.8069 | - | - | 0.3066 | 0.7551 | - | - | 0.3216 | 0.7420 | - | - |
| EPS3D (ours) | **0.6314** | **0.8546** | **0.4602** | **0.4044** | **0.6169** | **0.8469** | **0.3912** | **0.4329** | **0.4999** | **0.8629** | **0.3382** | **0.3232** | **0.4833** | **0.8512** | **0.3468** | **0.3106** |

the same 3D object should share similar semantics. To this end, we propose an Ins2Sem enhancement module. Specifically, for each training iteration, we randomly select $M$ Gaussian points as anchor points. For each anchor Gaussian $\mathcal{G}_m$, we select the top-$K$ neighboring Gaussians based on instance feature similarity, which are assumed to belong to the same 3D instance. We enforce semantic consistency among these Gaussians by aligning their semantic features. The following loss encourages the semantic features of Gaussians within the same instance to be consistent:

$$\mathcal{L}_{\text{Ins2Sem}} = \frac{1}{K} \frac{1}{M} \sum_{m=1}^{M} \sum_{k=1}^{K} (1 - \frac{S_k^m \cdot S_m}{\|S_k^m\|\|S_m\|}). \quad (7)$$

**Total loss objective.** In summary, the total loss objective for EPS3D is:

$$\mathcal{L}_{\text{total}} = w_1 \mathcal{L}_{\text{rgb}} + w_2 \mathcal{L}_{\text{ins}} + w_3 \mathcal{L}_{\text{sem}} + w_4 \mathcal{L}_{\text{Ins2Sem}}, \quad (8)$$

where $\mathcal{L}_{rgb}$ is the L1 loss for RGB rendering loss, following the common practices (Jiang et al., 2025; Sun et al., 2025; Fan et al., 2024) and $w_1, w_2, w_3, w_4$ are the weights for each separate loss term.

### 3.4.3. 3D PANOPTIC SEGMENTATION INFERENCE

For semantic segmentation, the final segmentation is achieved by calculating the cosine similarity between ren-dered pixel-wise semantic features and a collection of prototypes derived from text queries. Specifically, given a set of text prompts representing the target categories (e.g., "wall," "chair," "sofa"), the CLIP (Radford et al., 2021) text encoder produces corresponding feature prototypes denoted as $\{S_l^{\text{txt}}\}$. Moreover, the semantic segmentation label map $L_S$ is then determined by selecting the category with the highest similarity score $L_S = \text{argmax}_l(S_l^{\text{txt}} \cdot \hat{S}')$.

For instance segmentation, we can export the instance segmentation mask by performing the clustering algorithm (e.g., HDBSCAN (McInnes et al., 2017)) based on the discriminative 3D instance feature. Specifically, we can obtain all the centroids $\{I_q^{\text{Proto}}\}$ and it represents the prototype feature for the 3D instances. Then, the instance segmentation label map can be derived by $L_I = \text{argmax}_q(\text{sim}(I_q^{\text{Proto}}, I))$, where sim is the adopted similarity kernel function for supervising instance features (i.e., the dot product operation).

## 4. Experiments

### 4.1. Experimental Setup

**Implementation details.** We train EPS3D on the standard ScanNet (Dai et al., 2017) and ScanNet++ (Yeshwanth et al., 2023) datasets on eight NVIDIA A800 GPUs. We set the loss weights as $w_1 = 10^{-1}$, $w_2 = 10^{-3}$, $w3 = 10^{-1}$, and

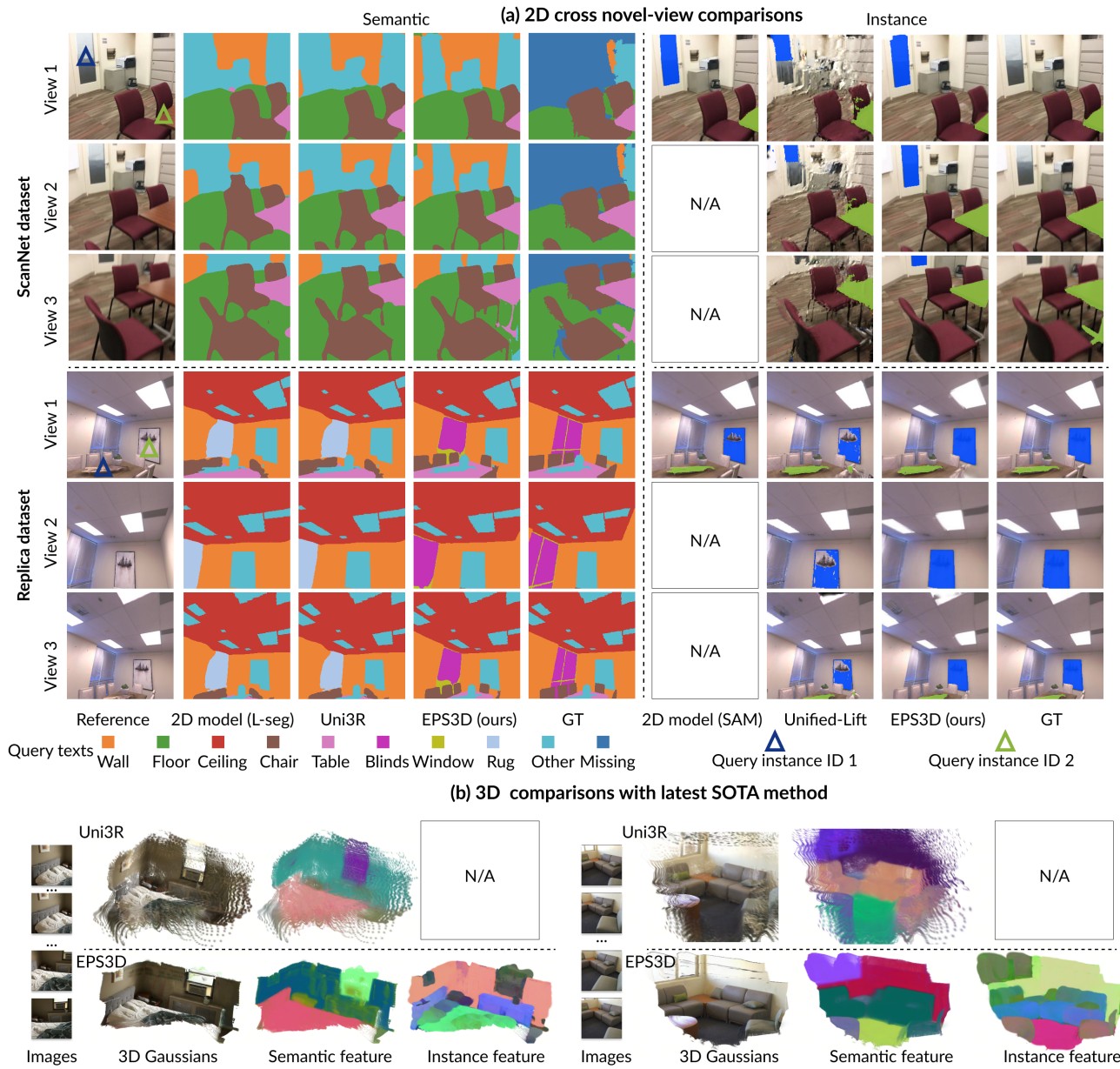

*Figure 4.* (a) Visual comparisons of novel views between our EPS3D and baselines on ScanNet (Dai et al., 2017) and Replica (Straub et al., 2019). For semantic understanding, we directly visualize the semantic segmentation maps according to the text queries across views. For instance-level understanding, we use the first novel view to select the 3D segmentation ID and visualize the corresponding segmentation across different views. We mark 'N/A' to indicate that the method does not support such predictions. Note that ScanNet annotations are incomplete in certain regions. (b) 3D visual comparisons between EPS3D and the SOTA baseline (i.e., Uni3R (Sun et al., 2025)).

$w_4 = 10^{-4}$ for the total loss function. For more details, please refer to the appendix.

**Evaluation benchmark and metrics** We conduct the experiments on the standard ScanNet (Dai et al., 2017) dataset and the Replica (Straub et al., 2019) dataset. For ScanNet dataset, we follow the common practice (Fan et al., 2024; Sun et al., 2025) to select the same test split with the semantic and instance annotation for evaluation. For Replica dataset, it contains 8 scenes with semantics and instance an-

notations for the evaluation. Specifically, we render the consistent semantic and instance mask for evaluation, following the protocol in LSM (Fan et al., 2024) and Unified-Lift (Zhu et al., 2025). Specifically, open-vocabulary segmentation is assessed via mean Intersection-over-Union (mIoU) and mean pixel accuracy (Acc), following the same evaluation protocol in LSM (Fan et al., 2024). As for instance segmentation, we first align predictions with ground truth using the linear assignment algorithm based on Intersection-over-

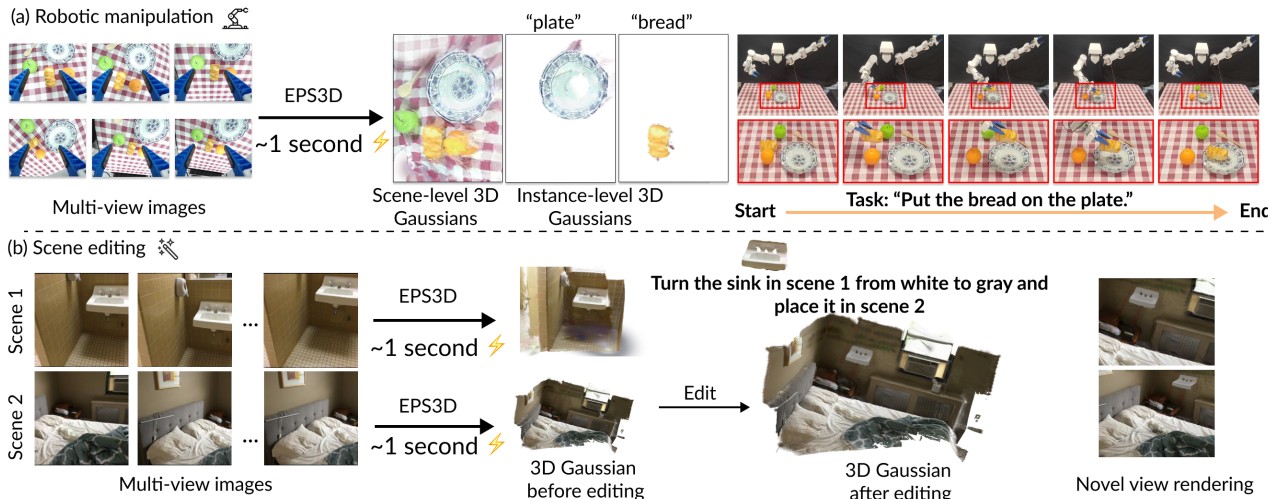

*Figure 5.* (a) EPS3D provides effective 3D panoptic segmentation with high efficiency, offering foundational 3D perception for robotic manipulation tasks. (b) Our EPS3D can recover both scene-level and instance-level Gaussians, which facilitates 3D scene editing (e.g., "Turn the sink in scene 1 from white to gray and place it in scene 2").

Union. We then report both mIoU and F-score, using an IoU threshold of 0.5, following Unified-Lift (Zhu et al., 2025).

**Baselines.** Our method focuses on the 3D end-to-end panoptic segmentation from multi-view images. We compare our method with the 2D open-vocabulary semantic foundation model (L-seg (Li et al., 2022)) and recent baseline (NeRF-DFF (Kobayashi et al., 2022), Feature-3DGS (Zhou et al., 2024), LSM (Fan et al., 2024) and Uni3R (Sun et al., 2025)) for the semantic part and compare our method with the 2D foundation model (SAM (Kirillov et al., 2023)) and recent 3D segmentation method Unified-Lift (Zhu et al., 2025) for the instance part.

### 4.2. Main Results

**Two-view setting.** We first evaluate our method with baselines on the standard indoor dataset for the 2-view setting, following the standard evaluation protocol as in LSM (Fan et al., 2024). As shown in the Tab. 1, our method not only achieve improved performance than the 2D models but also outperform then the recent SOTA baseline in terms of both semantic and instance segmentation metrics, including the optimization-based methods (e.g., NeRF-DDF (Kobayashi et al., 2022), Unified-Lift (Zhu et al., 2025)) and the recent feed-forward methods (i.e., LSM (Fan et al., 2024) and Uni3R (Sun et al., 2025)). Besides, we provide the time results that demonstrate that our method achieves significantly improved performance compared to the recent baseline while maintaining fast inference times.

**Multi-view setting.** To further evaluate the effectiveness of our method, we provide comparisons under the multi-view (i.e., 8 views) setting. As shown in Tab. 2, our method

*Table 3.* Panoptic metrics comparison on ScanNet (Dai et al., 2017) and on Replica (Straub et al., 2019). We follow the same experiment setting as Tab. 2.

| Method | ScanNet (Novel-view) | | | Replica (Novel-view) | | |
|---|---|---|---|---|---|---|
| | PQ ↑ | SQ ↑ | RQ ↑ | PQ ↑ | SQ ↑ | RQ ↑ |
| LSeg + SAM | 0.3803 | 0.4810 | 0.5127 | 0.2617 | 0.3682 | 0.2631 |
| Uni3R + Unified-Lift | 0.4013 | 0.5022 | 0.5189 | 0.2716 | 0.3751 | 0.2834 |
| EPS3D (ours) | **0.5304** | **0.6643** | **0.6759** | **0.3539** | **0.4925** | **0.4166** |

consistently achieves better performance for semantic and instance segmentation than the existing baselines. Moreover, the visual comparisons provided in Fig. 4 (a) further verify that our method supports consistent semantic and instance segmentation across views. Further, we provide the 3D visual comparisons between our method and the most recent baseline (Sun et al., 2025) in Fig. 4 (b), the results demonstrate our more discriminative 3D features.

### 4.3. More Analysis

**Complete panoptic evaluation.** We further report panoptic quality metrics (PQ, SQ, and RQ) to evaluate the complete panoptic segmentation performance. Since existing 3D methods focus on either semantic or instance segmentation, we construct two ensemble baselines: (i) combining 2D foundation models (LSeg + SAM), and (ii) combining the SOTA 3D methods (Uni3R for semantics + Unified-Lift for instance). As Tab. 3 reports, our method consistently outperforms both ensemble baselines across all panoptic metrics on both ScanNet and Replica, demonstrating the advantage of our unified panoptic prediction.

**Ablation study.** We provide an ablation study to analyze the effectiveness of the proposed technique components. Firstly, we observe that the supervision based on feature

*Table 4.* Ablation study on Replica (Straub et al., 2019). "Replace with Cross-attention" replaces our semantic-instance mutual enhancement module with standard cross-attention between semantic and instance features.

| Method | Novel Semantic | | Novel Instance | |
|---|---|---|---|---|
| | mIoU ↑ | Acc. ↑ | mIoU ↑ | F-sc. ↑ |
| EPS3D full pipeline | 0.4833 | 0.8512 | 0.3468 | 0.3106 |
| Without feature splatting | 0.4533 | 0.8112 | 0.2519 | 0.1510 |
| Without Ins2Sem module | 0.4531 | 0.8201 | 0.3388 | 0.3103 |
| Without Sem2Ins module | 0.4821 | 0.8500 | 0.3210 | 0.3017 |
| Replace with Cross-attention | 0.4677 | 0.8321 | 0.3230 | 0.3065 |

*Table 5.* Ablation on feature splatting for EPS3D on Replica (Straub et al., 2019). The evaluation is conducted under the context-view setting. ✓: feature splatting; ✗: DPT prediction.

| Method | Feature splatting | | Context Semantic | | Context Instance | |
|---|---|---|---|---|---|---|
| | Training | Inference | mIoU ↑ | Acc. ↑ | mIoU ↑ | F-sc. ↑ |
| Model 1 (Full) | ✓ | ✓ | **0.4999** | **0.8629** | **0.3382** | **0.3232** |
| Model 2 | ✓ | ✗ | 0.4965 | 0.8617 | 0.3365 | 0.3201 |
| Model 3 | ✗ | ✓ | 0.4615 | 0.8167 | 0.2638 | 0.1598 |
| Model 4 | ✗ | ✗ | 0.4606 | 0.8149 | 0.2609 | 0.1576 |

splatting is essential in our end-to-end 3D panoptic segmentation framework, and directly supervising the predicted feature from the semantic head and the instance head leads to a significant performance drop; see Tab. 4. Moreover, we further observe that removing each component in our semantic-instance mutual enhancement module will lead to suboptimal performance. Additionally, we ablate our semantic-instance mutual enhancement module by replacing it with standard cross-attention between semantic and instance features. As Tab. 4 reports, our design outperforms the cross-attention variant, showing the effectiveness of the proposed mutual enhancement module.

**Why does single-view supervision achieve view-consistent prediction?** We attribute the key factor that enables view-consistent predictions under single-view supervision to feature splatting. With feature splatting, the single-view supervision in EPS3D is no longer independent per-view teacher matching, but rather an implicit multi-view consistency objective. The ablation results in Tab. 4 verify the importance of feature splatting. To further disentangle its effects during training and inference, respectively, we conduct controlled experiments evaluated under the context-view setting. As Tab. 5 shows, feature splatting during training significantly enhances view-consistency, even without feature splatting at inference time (Model 2 vs. Model 1), which indicates that EPS3D has inherently learned to predict view-consistent 3D features. Specifically, during training with feature splatting, the loss gradient at each rendered pixel propagates to all Gaussians with non-zero splatting weights. Consequently, the single-view supervision with feature splatting encourages multi-view consistency.

*Table 6.* Analysis of different 2D teacher models on Replica (Straub et al., 2019).

| Method | Teacher Model | | Novel Semantic | | Novel Instance | |
|---|---|---|---|---|---|---|
| | Instance | Semantic | mIoU ↑ | Acc. ↑ | mIoU ↑ | F-sc. ↑ |
| Model 1 (default) | SAM | LSeg | 0.4833 | 0.8512 | 0.3468 | 0.3106 |
| Model 2 | Semantic-SAM | LSeg | 0.4841 | 0.8520 | 0.3485 | 0.3198 |
| Model 3 | SAM | MaskCLIP | 0.4788 | 0.8489 | 0.3460 | 0.3100 |

**Analysis on different 2D teacher models.** Following recent works (Fan et al., 2024; Zhu et al., 2025), we choose SAM (Kirillov et al., 2023) and LSeg (Li et al., 2022) as the default teacher models for a fair comparison. However, EPS3D supports other 2D models for supervision. We conduct experiments by replacing the instance teacher (SAM vs. Semantic-SAM (Li et al., 2023)) and the semantic teacher (LSeg vs. MaskCLIP (Zhou et al., 2022)), respectively. As Tab. 6 shows, using different teacher models provides comparable performance, demonstrating the generalizability of our framework to different 2D teacher models.

### 4.4. Application Results

Moreover, we further conduct the downstreaming application experiments to demonstrate the effectiveness of our method for the downstream applications.

**Robotic manipulation.** We showcase a robotic manipulation demo. Specifically, we leverage EPS3D to provide effective and efficient 3D reconstruction and perception for robotic manipulation tasks, building upon prior work (Zheng et al., 2024; Rashid et al., 2023; Ma et al.; Chen et al., 2026). As illustrated in Fig. 5 (a), the accurate segmentation produced by our method assists the robotic arm in completing grasping operations. More details are given in the appendix.

**3D scene editing.** Based on the reconstructed 3D panoptic Gaussians and segmented instance-level Gaussians, we can further perform 3D scene editing to facilitate applications in AR/VR. As demonstrated in Fig. 5 (b), based on our reconstructed Gaussian and segmentation results, we can easily apply recoloring effects and insert new objects into the original reconstructed 3D Gaussians, enabling novel view rendering for the edited 3D scene.

## 5. Conclusion

We propose EPS3D, a novel end-to-end 3D panoptic segmentation framework using multi-view images. Bypassing the need for 2D model preprocessing, EPS3D enables effective, multi-view consistent semantic and instance prediction in an end-to-end manner by distilling knowledge from 2D foundation models. We further propose a novel semantic-instance mutual enhancement module to encourage consistency between semantic and instance understanding, achieving significantly improved performance over existing SOTA baselines while maintaining high efficiency, and demonstrating strong potential for downstream applications.

## Acknowledgements

This study was supported by the InnoHK initiative of the Innovation and Technology Commission of the Hong Kong Special Administrative Region Government via the Hong Kong Centre for Logistics Robotics, HK RGC AoE under AoE/E-407/24-N.

## Impact Statement

The paper presents a novel end-to-end framework for open-vocabulary 3D panoptic segmentation, aiming to predict accurate semantic and instance features efficiently and effectively. By removing dependence on error-prone 2D preprocessing and enforcing semantic–instance consistency, the method improves robustness, efficiency, and scalability of 3D scene understanding. This approach supports practical machine learning applications that need robust 3D scene understanding, including robotic manipulation and navigation, augmented reality and virtual environment editing, embodied AI systems, and autonomous inspection. These capabilities can enhance real-world performance in settings such as healthcare assistance and industrial automation by enabling machines to perceive and act within complex spatial environments more reliably.

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

In this appendix, we further provide implementation details and more results.

## A. Implementation Details

**Detailed architecture.**  For the geometry transformer, inspired by (Wang et al., 2025a; Jiang et al., 2025), we first patchify images $C_i$ into $l_C = \frac{HW}{p^2}$ tokens of dimension $d$, where $p = 14$ and $d = 1024$, H, W are image height and width, respectively. To the resulting token sequence $t \in \mathbb{R}^{l_C \times d}$, we append a learnable camera token $t_g^i \in \mathbb{R}^{1 \times d}$ and four register tokens $t_R^i \in \mathbb{R}^{4 \times d}$. We omit the positional encodings on these additional tokens solely for the first view. The aggregated tokens $[t^i; t_g^i; t_R^i]$ across all $N$ views are fed into an $L$-layer Alternating-Attention transformer. In this architecture, each layer sequentially applies frame attention (operating on $\mathbb{R}^{N \times (l_C+5) \times d}$) and global attention (operating jointly over all views as $\mathbb{R}^{1 \times N(l_C+5) \times d}$). For the Gaussian head, we utilize the 3D-aware tokens to predict the standard 3D Gaussian properties. First, to establish the coordinate system, the refined camera tokens $\hat{t}_g^i$ are processed by a camera decoder $F_C$ (consisting of four self-attention layers and a linear head) to predict the camera parameters $p_i$. We anchor the global coordinate frame by setting the first camera pose to the identity transformation. With the camera poses established, we employ a dual-head architecture based on the DPT decoder (Ranftl et al., 2021) to regress the Gaussian primitives. The depth head $F_D$ processes image tokens $\hat{t}_i^I$ to generate the per-pixel depth maps $D_i$. Crucially, these predicted depths are back-projected via the estimated poses $p_i$ to determine the 3D positions (centers) $\{\mu_g\}_{g=1}^G$ of the Gaussians in the shared local coordinate frame defined above. Simultaneously, the Gaussian head $F_G$ combines the DPT features $F_d(\hat{t}_i)$ with the shallow CNN features $F_a(C)$ to predict the remaining attributes: opacity $\sigma_g$, orientation $r_g$, scale $s_g$, SH color coefficients $c_g$. The process is formalized as $D_i = F_D(\hat{t}^i)$, $\{\mu_g\} = \text{proj}(\{p_i\}, \{D_i\})$, $\{\sigma_g, r_g, s_g, c_g\} = F_b(F_d^2(\{\hat{t}^i\}) + F_a(\{C_i\}))$. For the semantic and instance heads, we utilize 3D-aware tokens to predict the corresponding features for each Gaussian. In the semantic branch, addressing the high dimensionality (i.e., 512) of the original CLIP features, the DPT layer first predicts a compressed feature vector (i.e., $\mathbb{R}^{32}$) to ensure memory-efficient rendering. We subsequently restore the high-dimensional features via projection layers, following common practices (Jiang et al., 2025; Sun et al., 2025). For the instance branch, we directly regress the instance features at their final dimension (i.e., 32). Finally, to promote the compactness of the Gaussian primitives, we employ the voxelization procedure (Jiang et al., 2025) to obtain voxel-averaged panoptic Gaussians for rendering.

**Training details.**  For fair comparison, we train EPS3D on the standard ScanNet (Dai et al., 2017) and ScanNet++ (Yeshwanth et al., 2023) datasets, following common pratice (Fan et al., 2024; Sun et al., 2025). The transformer layers, camera head, and depth head are initialized with weights from the pretrained Anysplat (Jiang et al., 2025), while the semantic and instance heads are randomly initialized. During training, input images are resized to a maximum long-edge resolution of 518 pixels and augmented via random horizontal flipping. In each iteration, we define a window size of $N = 24$ and randomly sample $N_c$ input views such that $2 \leq N_c \leq N$. Optimization is performed using AdamW with a cosine learning rate schedule starting at $2e-4$. To ensure training stability, we freeze the geometry transformer backbone and jointly optimize the Gaussian head, semantic head, and instance features. We set the loss weights as $w_1 = 1e-1$, $w_2 = 1e-3$, $w3 = 1e-1$, and $w_4 = 1e-4$ for the total loss function. To ensure more effective basic Gaussian training, we also adopt the geometry constraint loss, following the practice in Anysplat (Jiang et al., 2025). EPS3D is trained on eight NVIDIA A800 GPUs for approximately two days, with the batch size set to 1 for each GPU. For the Ins2Sem module, we select $M = 1000$ anchor Gaussians to balance efficiency and coverage, and $K = 30$ neighbors empirically.

**Experimental details**  We conduct our evaluation on the standard ScanNet and Replica datasets. For ScanNet, we utilize the version preprocessed by LSM (Fan et al., 2024), which comprises 40 scenes. For Replica, we employ the complete set of 8 scenes. Following the semantic evaluation protocol of LSM (Fan et al., 2024), we use the standard text queries: wall, floor, ceiling, chair, table, bed, sofa, others to assess semantic segmentation performance (i.e., mIoU, Acc) on ScanNet. Similarly, for the Replica dataset, we use the text queries wall, ceiling, floor, chair, blinds, sofa, table, rug, window for evaluation (i.e., mIoU, Acc). To evaluate instance segmentation, we first randomly sample 50,000 Gaussians and apply HDBSCAN clustering based on their instance features. We then adopt the evaluation protocol from Unified-Lift (Zhu et al., 2025) to compute the final metrics (i.e., mIoU, F-score) for ScanNet and Replica datasets. For a fair comparison with the test-time methods (Feature-3DGS (Zhou et al., 2024) and Unified-Lift (Zhu et al., 2025)), we employ VGGT (Wang et al., 2025a) to pre-process the scenes, producing point clouds and camera poses as initialization. This allows us to avoid potential failures associated with relying on COLMAP. For the test-time baselines, we train each model for 5000 iterations.

**Robotic application** To demonstrate the real-world applicability of EPS3D, we implement a manipulation pipeline inspired by recent advancements (Rashid et al., 2023; Zheng et al., 2024; Huang et al., 2024b). For a high-level command such as "put the bread on the plate", we leverage Qwen2-VL (Wang et al., 2024a) to decompose the instruction and output text descriptions of the target objects (e.g., bread" and "plate"). Then, we can utilize EPS3D to conduct the instance-level segmentation and we can utilize Qwen2-VL to select the segmentation ID from a reference view, producing the corresponding instance-level Gaussians. These segmented Gaussians subsequently are used to estimate the grasp poses (Mousavian et al., 2019) and manipulation parameters (Tang et al., 2025; Huang et al., 2024b).

## B. More Results

We provide more 3D visual comparisons with the latest SOTA methods in Fig. 6. We also provide visual comparisons with broader baselines (LSM (Fan et al., 2024), Feature-3DGS (Zhou et al., 2024)) in Fig. 7. The results consistently demonstrate that our method provides more accurate and consistent segmentation with fewer artifacts.

## C. Limitations

In this work, we focus on static indoor scenes and do not address dynamic environments, where objects or agents may move over time. Effectively extending the framework to handle dynamic scenarios remains an open question and requires further effort. Additionally, the performance and generalization capability of our model are still constrained by the diversity of indoor datasets. We believe that leveraging both indoor and outdoor datasets for training would further improve the robustness of our method. We plan to explore these directions in future work.

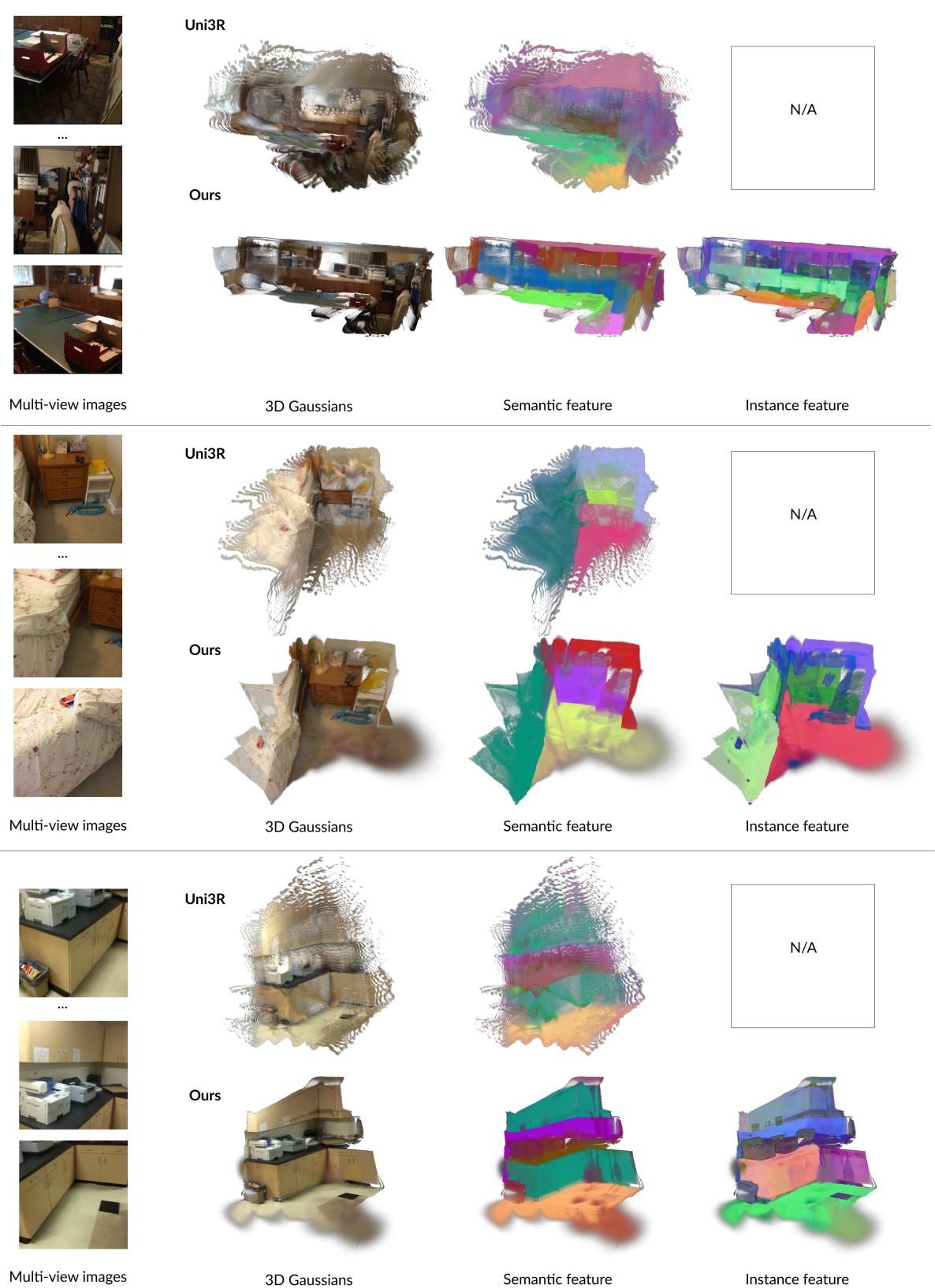

*Figure 6.* 3D comparisons between our method and the latest SOTA method (Sun et al., 2025). We mark 'N/A' to indicate that the method does not support such predictions.

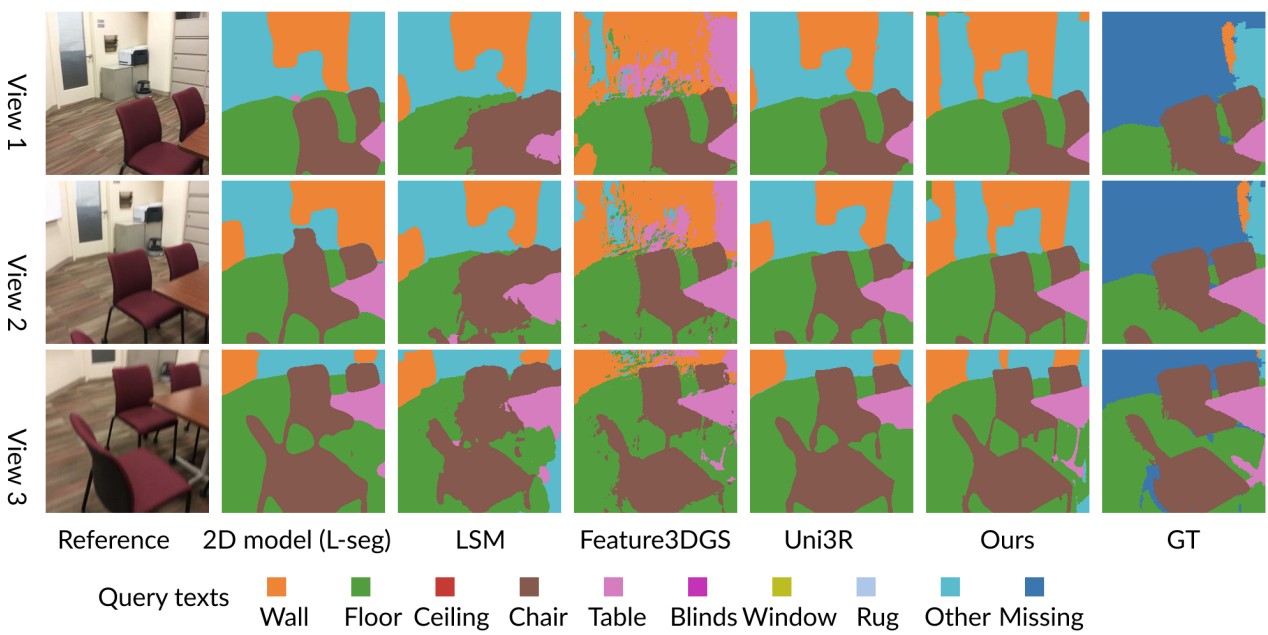

*Figure 7.* More visual comparisons with broader baselines (Feature-3DGS (Zhou et al., 2024), LSM (Fan et al., 2024)).

