# OpenReview forum: "EPS3D: End-to-End Feed-Forward 3D Panoptic Segmentation"
_ICML.cc/2026/Conference — ICML 2026 regular_

### Official Review · Reviewer_jEnx · 2026-03-10

**Soundness:** 3
**Presentation:** 2
**Significance:** 4
**Originality:** 3
**Overall Recommendation:** 4
**Confidence:** 4

**Summary:**

This paper proposes EPS3D, an end-to-end, feed-forward framework for open-vocabulary 3D panoptic understanding from multi-view images, built on a unified 3D Gaussian representation with semantic and instance features. I like the overall direction: avoiding per-scene optimization and aiming for fast, view-consistent 3D predictions is valuable for practical 3D perception and downstream applications. The method is generally well-motivated and the results look promising.

**Compliance With Llm Reviewing Policy:**

Affirmed.

**Final Justification:**

Thank you for the author's rebuttal which dispelled my concerns, so I have maintained a positive rating.

**Key Questions For Authors:**

see weakenss

**Limitations:**

Exploration on robustness of diverse scene and dynamic 3D Guassian

**Strengths And Weaknesses:**

### Strengths

1. An end-to-end, feed-forward OV3D pipeline that avoids heavy multi-stage fusion.
2. A clean unified 3D Gaussian representation that jointly encodes geometry plus semantic and instance features.
3. Strong performance and practical value in robotics and interactive applications.

### Weaknesses

1. There’s a bit of a gap between the “panoptic” claim and what’s actually shown in the paper. The results are mostly reported as semantic and instance numbers side-by-side, but I didn’t see a clear, complete panoptic evaluation. In typical panoptic segmentation setups, it’s common to report PQ / SQ / RQ, and that part seems missing here.
2. The training heavily relies on distillation from 2D foundation models. The paper argues it fixes view inconsistency, but the supervision from LSeg and SAM is still per-view. So I’m not fully convinced the end-to-end training is truly enforcing rendered cross-view consistency, rather than mainly learning to match view-dependent teacher signals. A more direct analysis or controlled experiment would help clarify this.
3. Some key method details and robustness experiments are under-explored. For example, how are N (anchors) and K (neighbors) selected, and how sensitive are the results to these choices? Also, since the whole approach depends on 2D teachers, it would be important to show how performance changes with different foundation model choices, and whether the method remains stable under different settings or more challenging conditions.

---

> ### Author Rebuttal · Authors · 2026-03-31
>
> We thank Reviewer jEnx for the positive and insightful review. We appreciate that the reviewer values the overall direction of avoiding per-scene optimization for fast, view-consistent 3D prediction, recognizes the clean unified 3D Gaussian representation, and sees strong practical value in the method. Below we address each concern.
>
> ---
>
> ## Q1: Complete panoptic evaluation.
>
> Thanks for the great suggestion. Following the reviewer’s suggestion, we report PQ, SQ, and RQ. Since existing 3D methods focus on either semantic or instance segmentation, we therefore construct two ensemble baselines: (1) combining 2D foundation models (LSeg + SAM), and (2) combining the strongest 3D methods (Uni3R for semantics + Unified-Lift for instance). As shown in [🔗📊**Table A**](https://anonymous.4open.science/r/anonymous-8F7E/Reviewer_jEnx_Tab_PQ_metric.png), EPS3D consistently outperforms both ensemble baselines across all panoptic metrics, demonstrating the advantage of unified panoptic prediction.
>
> ---
>
> ## Q2: Why single-view supervision can achieve view-consistent predictions.
>
> Thanks for your insightful question. In the main paper ablation study (see Tab. 3, main paper), we identify that one key factor for achieving multi-view consistency under single-view supervision is feature splatting during training. Particularly, with feature splatting, the single-view supervision is no longer independent per-view teacher matching, but rather an implicit multi-view consistency objective.
>
> To better verify this, we further conducted controlled experiments to isolate the effects of feature splatting during training and inference separately, specifically evaluated under the context-view setting. As shown in [🔗📊**Tab. B**](https://anonymous.4open.science/r/anonymous-8F7E/Reviewer_jEnx_Tab_feature%20splatting.png), the feature splatting during training time significantly enhances the view-consistency, even without feature splatting at inference time (Model 2 vs. Model 1), which indicates that proposed EPS3D has inherently learned to predict view-consistent 3D features. Specifically, during training time with feature splatting, the loss gradient at each rendered pixel propagates to all Gaussians with non-zero splatting weights. Consequently, the single-view supervision with feature splatting implicitly encourages multi-view consistency.
>
> ---
>
> ## Q3: Hyper-parameter details.
>
> Regarding the hyperparameters in the Ins2Sem module, we selected $N=1000$ for anchors to balance the efficiency and coverage, and $K=30$ for neighbors empirically. We further perform the sensitivity experiment. The results are reported in [🔗📊**Tab. C**](https://anonymous.4open.science/r/anonymous-8F7E/Reviewer_jEnx_Tab_hyperparamter.png), showing that the model’s performance remains highly robust across varied $N$ (500-2000) and $K$ (15-60) values.
>
> ---
>
> ## Q4: Difference to 2D teacher models.
>
> Thanks for your suggestion. In our main paper, following recent works [1,2], we choose SAM and LSeg as teacher models for a fair comparison. However, proposed EPS3D supports other 2D models for supervision. Here, we conduct experiments by replacing the instance teacher (SAM → Semantic-SAM [3]) and the semantic teacher (LSeg → MaskCLIP [4]) respectively.  Results in [🔗📊**Tab. D**](https://anonymous.4open.science/r/anonymous-8F7E/Reviewer_jEnx_Tab_Teacher_models.png) show that using different teacher models provides comparable performance. We will add more experiment results with more discussion in the revised version.
>
>
>
> ---
>
> ## Q5: Stabilities in more diverse settings.
>
> Thanks for your suggestion. In main paper, we evaluate the proposed EPS3D on indoor benchmarks following the latest baselines[1,2], and additionally test EPS3D in the robotics demo. Here, we further provide the visual result for the outdoor scene. As shown in [🔗🖼️**Fig. A**](https://anonymous.4open.science/r/anonymous-8F7E/Reviewer_jEnx_Fig_Outdoor.png), EPS3D manages to provide a consistent and accurate prediction. We agree it remains an open question to improve generalization to more diverse scene settings and we leave the comprehensive exploration for future work.
>
> [1] Large Spatial Model: End-to-end Unposed Images to Semantic 3D. NeurIPS 2024.
> [2] Unified 3D Reconstruction and Semantic Understanding via Generalizable Gaussian Splatting from Unposed Multi-View Images. CVPR 2026.
> [3] Semantic-SAM: Segment and Recognize Anything at Any Granularity. ECCV 2024.
> [4] Extract Free Dense Labels from CLIP. ECCV 2022.

---

> > ### Author Rebuttal · Reviewer_jEnx · 2026-04-01
> >
> > Thanks for the detailed rebuttal and for sharing the additional figures and tables through the anonymous link. The new explanations and experimental results cleared up my confusion, and I’ll keep my positive assessment.

---

> > > ### Author Response · Authors · 2026-04-03
> > >
> > > Dear Reviewer jEnx,
> > >
> > > Thank you for your feedback on our rebuttal. We are glad to know that our response has addressed your questions. We greatly appreciate your kind support for our work!
> > >
> > > Sincerely,
> > > The Authors

---

### Official Review · Reviewer_785z · 2026-03-13

**Soundness:** 3
**Presentation:** 3
**Significance:** 2
**Originality:** 2
**Overall Recommendation:** 4
**Confidence:** 3

**Summary:**

EPS3D is an end-to-end feed-forward 3D panoptic segmentation framework that takes multi-view RGB images as input and directly predicts a unified 3D Gaussian representation encoding geometry, appearance, semantics, and instance identity, with an inference speed of ~1 second per scene. During training, CLIP and SAM serve as teacher models for distillation supervision, and a semantic-instance mutual enhancement module (Ins2Sem/Sem2Ins) is introduced to encourage consistency between the two feature branches. The method outperforms existing baselines on ScanNet and Replica benchmarks, and supports downstream applications such as robotic grasping and 3D scene editing.

**Compliance With Llm Reviewing Policy:**

Affirmed.

**Final Justification:**

I would like to thank the authors for their comprehensive and convincing rebuttal. The authors have addressed all of my concerns. I am raising my score to "Weak Accept".

**Key Questions For Authors:**

I encourage the authors to carefully address the concerns raised in the weaknesses section, as these issues are central to my evaluation of the paper.

**Limitations:**

See the weaknesses

**Strengths And Weaknesses:**

Strengths
- The problem setting is meaningful and practically relevant. Open-vocabulary 3D panoptic segmentation is a more complete and useful problem than standard 3D semantic understanding, and it is clearly closer to downstream applications such as robotics and scene editing.

- The overall method is clear and easy to follow. The paper has a clean main story, and the roles of the main components are easy to understand. The semantic-instance interaction is also intuitively motivated.

- The empirical results are overall strong.The paper provides fairly comprehensive comparisons, and the method shows strong performance across multiple settings.

Weaknesses:
- Limited technical novelty. Most of the framework is built from existing components, including a VGGT-style backbone, DPT heads, and 3DGS. The only clearly new pieces are Sem2Ins and Ins2Sem. These modules are not unreasonable, but they feel more like a fairly natural way to couple semantic and instance features than a methodological contribution. The gains they bring are also not strong enough to make the novelty particularly convincing.

- Insufficient ablation and analysis. The ablation study is very limited and is only conducted on Replica, which has just 8 scenes. That is not enough to justify the proposed design. Important factors such as the hyperparameters in Ins2Sem, the loss weights, and even the choice of prediction head are not analyzed. More importantly, the paper never really shows why this particular semantic-instance interaction design is the right one. There is no comparison with more standard alternatives such as cross-attention or other feature fusion schemes.

- The claim about “bypassing 2D preprocessing” is overstated. The method does avoid running a separate 2D model at inference time for preprocessing, but during training it still relies heavily on supervision from 2D foundation models, including distillation targets and masks. So the dependence on 2D models is not removed; it is simply shifted from inference time to training time. That distinction matters, and the paper should state it more carefully.

---

> ### Author Rebuttal · Authors · 2026-03-31
>
> We thank Reviewer 785z for the detailed evaluation. We appreciate the reviewer’s recognition that the problem setting is meaningful and practically relevant, that the method is clear to follow, and that the empirical results are strong. We also take seriously the reviewer’s concerns and address them below.
>
> ---
>
> ## Q1: Core contributions.
>
> Thanks for raising this point. We acknowledge that the proposed EPS3D is built on existing components that have proven effective across various tasks. However, the primary contributions of our work lie in two key aspects:
>
> First, we propose the **first end-to-end feed-forward** framework for open-vocabulary 3D panoptic segmentation; this unified formulation significantly reduces error accumulation from the independently inferred 2D results used as preprocessing in existing works (see main paper Fig.2). Extensive experiments show that EPS3D achieves superior performance (e.g., +13% mIoU for semantics on Replica) for view-consistent segmentation within a unified model.
>
> Second, instead of **solely learning either semantic or instance prediction in existing methods**, we propose to **jointly learn the two complementary features to provide comprehensive open-vocabulary panoptic understanding**. Building on this insight, we design the **tailored** semantic-instance mutual enhancement, learning to produce coherent 3D panoptic results. We have further conducted comparisons with the suggested alternative (i.e., standard cross-attention) and experiments verify that such a tailored design is more effective. Please refer to Q3.4 for more analysis.
>
> ---
>
> ## Q2: Clarification on “bypassing 2D preprocessing”
>
> Thanks. We would like to clarify a key distinction in EPS3D with respect to “bypassing 2D preprocessing”:
>
> - **Existing feedforward methods (Uni3R, LSM):** They adopt a **two-stage** solution. First, these methods rely on additional preprocessing that utilizes 2D models to independently infer predictions per view. Subsequently, the 2D results are used as the input for 3D fusion, which often leads to cross-view inconsistencies and error accumulation.
> - **EPS3D:** Instead of this paradigm, we adopt an **end-to-end framework** where segmentation is **directly** predicted across views for both training and inference, with 2D models serving exclusively as training-time teachers rather than as intermediate inputs to the 3D network.
>
> Thus, we believe this is an architectural distinction, more than a simple timing shift. Moreover, it brings two benefits: (1) **Better performance**: as shown in Tab. 1 and Tab. 2 of main paper, the end-to-end framework inherently learns to predict view-consistent semantic and instance features, which consistently outperforms two-stage feed-forward methods across both datasets; (2) **Simpler deployment**: Once training is finished, EPS3D requires no external 2D foundation model, reducing system complexity for applications such as robotic manipulation.
>
> We will revise the phrasing to make the distinction between “2D preprocessing” and “training-time supervision” more explicit.
>
> ---
>
> ## Q3: More ablation and analysis.
>
> 1) ScanNet Ablations: We have expanded ablations to ScanNet, as shown in [🔗📊**Tab. A**](https://anonymous.4open.science/r/anonymous-8F7E/Reviewer_785z_Tab_Complete%20ablation.png), confirming the necessity of both Sem2Ins and Ins2Sem modules across different datasets.
>
> 2) Hyperparameters ($K$, $N$): We evaluated the sensitivity of N and K in Ins2Sem on both ScanNet and Replica. As shown in [🔗📊**Tab. B**](https://anonymous.4open.science/r/anonymous-8F7E/Reviewer_785z_Tab_hyperparameters.png), performance remains highly stable across (500-2000) for $N$  and (15-60) for $K$.
>
> 3) Prediction head: We adopt the DPT head following established practice in all recent feed-forward 3D methods [1,2,3,4]. This provides a controlled setting for evaluating the proposed EPS3D. We consider exploring alternative heads orthogonal to our contributions and leave it for future work.
>
> 4) Comparison with cross-attention alternative: Motivated by the asymmetric complementarity between semantic and instance features, we design semantic-instance mutual enhancement module for more coherent prediction. Specifically, Sem2Ins acts as a **conditioning** mechanism that provides category-level context for instance discrimination. Ins2Sem acts as a **regularization** mechanism that constrains semantics within each instance to be consistent. Together, they promote coherent panoptic prediction. These two directions serve distinct purposes — one facilitates discrimination, the other enforces consistency. Standard cross-attention treats both directions symmetrically and does not explicitly separate these two roles. The results in [🔗📊**Tab. C**](https://anonymous.4open.science/r/anonymous-8F7E/Reviewer_785z_Tab_Cross-attention.png) further confirm the design outperforms cross-attention.
>
> [1] VGGT, CVPR'25; [2] AnySplat, SIGGRAPH Asia'25; [3] LSM, NeurIPS'24; [4] Uni3R, CVPR'26.

---

> > ### Author Rebuttal · Reviewer_785z · 2026-04-03
> >
> > I would like to thank the authors for their comprehensive and convincing rebuttal. The authors have addressed all of my concerns. I am raising my score to "Weak Accept".

---

> > > ### Author Response · Authors · 2026-04-03
> > >
> > > Dear Reviewer 785z,
> > >
> > > Thank you for your feedback on our rebuttal. We are glad to know that our response has addressed your questions. We sincerely appreciate your kind support for our work!
> > >
> > > Sincerely,
> > > The Authors

---

### Official Review · Reviewer_S64H · 2026-03-13

**Soundness:** 3
**Presentation:** 3
**Significance:** 2
**Originality:** 2
**Overall Recommendation:** 4
**Confidence:** 3

**Summary:**

The paper shows a method for end-to-end feed-forward 3D panoptic segmentation. The paper extends VGGT and Anysplat capabilities by introducing additional components in the model for instance and semantic predictions. The paper also introduces specialized loss functions and additional techniques to preserve multiview consistency across predictions and prevent bleeding. Presented method EPS3D is evaluated across common datasets and compared with SOTA methods. Additionally, the paper shows real world applications of the following method.

**Compliance With Llm Reviewing Policy:**

Affirmed.

**Final Justification:**

I am satisfied with the rebuttal and, as indicated, I recommend this paper for weak accept

**Key Questions For Authors:**

I’m willing to increase my score if authors address the weaknesses of the paper and answer the following questions:

 1. VGGT and Anysplat are known to require substantial GPU memory. Since EPS3D builds upon these methods, it is likely that it also has significant GPU memory requirements. Please report the GPU memory usage during inference and provide a comparison with the other evaluated methods.
2. The paper demonstrates a robotic application. However, it is unclear whether the system was running directly on the physical robot or on a remote server. If the latter, did the setup require an internet connection? Clarifying the deployment setup would help better understand the practical applicability of the method.
3. EPS3D is compared quantitatively with many methods in the paper, but qualitatively with only four. The paper could benefit from providing more extensive qualitative comparison with more SOTA methods (NeRF-DFF, Feature-SDGS, LSM).
4. Methods such as SAM and LERF allow not only for prompting for the whole object but also for its subparts to be selected. Is this method also supporting that?

**Limitations:**

The paper demonstrates a robotic application; however, it is unclear whether the system was running directly on the physical robot or on a remote server. If the method requires substantial GPU resources, this may limit its applicability in real-world robotic systems, where onboard computational resources are often constrained. The authors may consider discussing the computational requirements and potential deployment limitations more explicitly.

**Strengths And Weaknesses:**

Strengths:
1. The paper is well written and introduces concepts clearly. It guides the reader through the problem of creating the panoptic segmentation that eliminates error accumulation, while jointly supporting accurate semantic and object-level predictions in 3D. Methodology is well structured and complemented with math equations.
2. The figures presented in the paper are clear and contain a lot of information and because they are very well structured, they are easy to understand. Additionally, the teaser image is well summarizing what the content of the paper is going to be.
3. The paper proves that the methods outperform other SOTA methods by a big margin. The inference time of around one second is competitive compared to other methods and appears suitable for many practical applications.

Weaknesses:

 1. The figure 3 could contain math symbols introduced in methodology section. This would make it easier to follow the pipeline while reading the text. My biggest concerns regard figure 4. GT images and EPS3D images in the “Semantic” part of the figure are blurry and contain a lot of (jpeg like) artifacts. This denies the sense of qualitative comparisons and has to be fixed. Moreover, the example from Replica dataset in the ''Instance'' part isn’t the best one. I understand that EPS3D correctly selects the whole window but it was prompted with the window part which is clearly shown on GT. This example is ambiguous in that manner, because one can argue that selecting the whole window is also correct. I’m aware of Replica not being perfectly annotated but the example in the paper shouldn’t contain such ambiguities and I would suggest replacing it. Finally in table 1 the Venue column is not adding any value to the paper content and I would suggest removing it.
2. The proposed approach incorporates several ideas from existing methods (VGGT and Anysplat), and the overall level of novelty seems somewhat incremental. The main novelty lies in the Sem2Ins and Ins2Sem components, which, as demonstrated by the ablation studies, contribute significantly to the strong performance. Nevertheless, while the paper introduces some novel elements, the overall methodological contribution remains relatively limited, as the improvements are largely achieved through relatively simple techniques.
3. Sem2Ins is introduced as a key component, bit its role in the overall system is not clearly explained. In particular, it is unclear how I^{sem}_{g} is used within the method. It appears that it may be related to Eq. (5), but this connection is not explicitly clarified in the paper and should be explained more clearly.

---

> ### Author Rebuttal · Authors · 2026-03-31
>
> We thank Reviewer S64H for the thorough and constructive review. We appreciate the reviewer’s recognition that the paper is well written, clearly presented, and supported by strong empirical results with practical relevance. Below we address each concern.
>
> ---
>
> ## Q1: Suggestions for paper writing.
>
> We appreciate the constructive feedback. We will update main paper Fig. 3 with explicit math symbols matching the methodology. Regarding main paper Fig. 4, we will use uncompressed images and update with more comprehensive qualitative examples with a more clearly-delineated object. Main paper Table 1’s “venue” column will be removed.
>
> ---
>
> ## Q2: Relationship between EPS3D and 3D reconstruction methods.
>
> We appreciate this question. While EPS3D builds upon VGGT and 3DGS as its geometric backbone, our contribution goes beyond assembling existing components in two aspects.
>
> First, we propose the **first end-to-end feed-forward** framework for open-vocabulary 3D panoptic segmentation; this unified formulation significantly reduces error accumulation from the independently inferred 2D results used as preprocessing in existing works (see Fig.2 in our paper). Extensive experiments manifest that EPS3D, as a unified model, achieves superior performance (e.g., **+13%** mIoU on Replica) for view-consistent segmentation with high efficiency (e.g., 1s per scene).
>
> Second, instead of **solely learning either semantic or instance prediction in existing methods**, we propose to **jointly learn the two complementary features** for comprehensive open-vocabulary panoptic segmentation. Building on this insight, we further design the tailored semantic-instance mutual enhancement, learning to provide more coherent 3D understanding. The ablation study is performed in the main paper to verify the effectiveness.
>
> ---
>
> ## Q3: More explicit explanation in the Sem2Ins paragraph.
>
> Thanks for pointing this out. The semantic-refined instance feature $I^{sem}_g$ (main paper Eq. 6) is used as the final instance attribute for the 3D Gaussians. When rendering the 2D instance map, this rendered feature is directly supervised by the instance contrastive loss (main paper Eq. 5). We will explicitly clarify this in the revision.
>
> ---
>
> ## Q4: GPU memory.
>
> We compare EPS3D with the latest SOTA method (i.e., Uni3R). Thanks to our unified end-to-end design, EPS3D requires no external 2D foundation model at inference, resulting in smaller GPU memory usage while achieving superior performance, as shown in the table below.
>
> Table A. GPU memory comparisons on the Replica dataset for multi-view (i.e., 8-view) image input. We report the semantic mIoU metrics for the novel-view setting.
>
> |  | GPU | mIoU |
> | --- | --- | --- |
> | Uni3R | 16.0 G | 0.3216 |
> | EPS3D| 11.0 G| 0.4833 |
>
>
> ---
>
> ## Q5: Robot platform and internet connection.
>
> For the robotic application, the whole system is deployed directly on a physical tabletop robotic platform equipped with a local computer using a single onboard A6000 GPU (48G), following a common setup [1,2,3] for modern robotic platform. **All models and computations are performed locally.** **No Internet** connection is required during the deployment and operation.
>
> [1] GaussianGrasper: 3D Language Gaussian Splatting for Open-vocabulary Robotic Grasping.
> [2] Language Embedded Radiance Fields for Zero-Shot Task-Oriented Grasping.
> [3] D3Fields: Dynamic 3D Descriptor Fields for Zero-Shot Generalizable Rearrangement.
>
> ---
>
> ## Q6: Visual results with broader baselines.
>
> Thanks for your suggestion. In the main paper, we mainly compare EPS3D with the strongest SOTA method. We provide visual comparisons with more baselines (LSM, Feature-3DGS) in [🔗🖼️**Fig. A**](https://anonymous.4open.science/r/anonymous-8F7E/Reviewer_S64H_Fig_More_visual_comparisons.png), and visual results consistently demonstrate that our method provides more accurate and consistent segmentation with fewer artifacts. We will update more visual results in the revised version.
>
> ---
>
> ## Q7: Extension to part segmentation.
>
> Thanks for your insightful question. In this paper, we follow the same object-level setting as the latest baselines [4,5,6] to provide direct comparisons, showing that EPS3D achieves superior performance. Meanwhile, our end-to-end framework can be extended to support part-level segmentation. To demonstrate this, we have conducted an extension experiment by introducing an additional “part feature head” supervised by the sub-part segmentation generated by SAM. As shown in [🔗🖼️**Fig. B**](https://anonymous.4open.science/r/anonymous-8F7E/Reviewer_S64H_Fig_Part-level.png), given a prompt, EPS3D can predict both instance-level segmentation (the whole chair) and part-level segmentation (the backrest only) consistently across views. We agree that part-level extension is an interesting direction and we leave deeper exploration for future work.
>
> [4] LSM... NeurIPS 2024.
> [5] Uni3R... CVPR 2026.
> [6] Unified Lift... CVPR 2025.

---

> > ### Author Rebuttal · Reviewer_S64H · 2026-04-01
> >
> > I would like to thank the Authors for addressing all the points I raised. I appreciate the clarifications regarding the equations, as well as the additional details provided about the robotic experiment and its setup. I am also glad to see that the Authors responded to my comments by including more examples in the figures. The additional experiment with instance-level segmentation looks promising and can be a great extension to the system. Overall, I am satisfied with the rebuttal and, as indicated, I recommend this paper for weak accept

---

> > > ### Author Response · Authors · 2026-04-03
> > >
> > > Dear Reviewer S64H,
> > >
> > > Thank you for your feedback on our rebuttal. We are pleased to know that our response has addressed your questions. We sincerely appreciate your kind support for our work!
> > >
> > > Sincerely,
> > > The Authors

---

### Decision · Program_Chairs · 2026-04-30

**Decision:**

Accept (regular)

**Comment:**

The paper proposed a feed forward model for 3D panoptic segmentation. The work initially received mixed feedback, with reviewers requesting clarification in terms of some key model characteristics (e.g. memory usage), experimental validation (e.g. ablations) and applications/use cases (e.g. in robotics). The rebuttal and author-reviewer discussion seems to have solved most of the initial concerns from the reviewers, leading to a consensus for acceptance (4,4,4). Based on the reviews, discussion and rebuttal the AC agrees with this consensus and recommends acceptance for this work - congratulations!